# The interplay between foraging choices and population growth dynamics

Jimmy Calvo-Monge[1], Baltazar Espinoza[2]*,
Fabio Sanchez[3], Jorge Arroyo-Esquivel[4]

**1** Centro de Investigación en Matemática Pura y Aplicada, Universidad de Costa Rica, Ciudad Universitaria Rodrigo Facio, San José, Costa Rica, **2** Biocomplexity Institute, University of Virginia, Virginia, United States of America, **3** Escuela de Matemática-CIMPA, Universidad de Costa Rica, Ciudad Universitaria Rodrigo Facio, San José, Costa Rica, **4** University of California Davis, California, United States of America

☙ These authors contributed equally to this work.
* baltazar.espinoza@virginia.edu

**Data availability statement:** All relevant data are within the manuscript and its Supporting Information files.

## Abstract

Population growth models typically incorporate attributes observable at the population scale, often overlooking the trade-off between individual-level reproductive and behavioral traits and their influence on population size. Individuals' survival and reproductive abilities are expected to dynamically evolve depending on the population size, which is affected by the aggregation of individual decisions. Reconciling individual-level incentives with population-level dynamics requires an integrative framework that explicitly addresses the intertwined relationships between population growth and individual decision-making processes. We formulate a multiscale modeling framework that integrates the logistic population growth model with an optimal foraging model to study the interplay between individual-level behavioral incentives and population growth dynamics. Specifically, we explicitly model individuals' decision-making process, which shapes their reproductive fitness and, ultimately, influences population growth. Moreover, we incorporate the concept of resource limitations from the logistic growth model to account for dynamic incentives that depend on population size. Our results yield insights into the multiscale processes, such as the selection pressure of behavioral choices and the cost-benefit of social activities that influence population robustness beyond mere size and aggregated reproductive traits. We found that populations exhibiting similar limiting sizes may undergo significantly different transient dynamics. This variation may be induced by environments imposing distinct behavioral cost-benefit trade-offs that require individuals to exert different levels of foraging effort to maintain reproductive viability.

## Introduction

The logistic population growth model is a fundamental framework in ecology, describing population growth under resource constraints. Unlike exponential growth, which assumes limitless resources, the logistic model incorporates the concept of carrying capacity—the maximum population size an environment can support. Originally developed by Pierre

**Funding:** National Science Foundation (NSF) through Expeditions in Computing Grant CCF-1918656. NSF through Incorporating Human Behavior in Epidemiological Models (IHBEM) grant DMS-2327710

**Competing interests:** The authors have declared that no competing interests exist.

François Verhulst in 1838 [1], the model was formulated to understand the dynamics of population growth better as it encounters environmental limits [2,3]. The logistic population growth is a pivotal concept due to its capacity to incorporate changes in systems constrained by limited resources, making it a valuable tool for effective decision-making. Historical work has demonstrated its applicability not only in ecology, where it emerged, supported by substantial empirical evidence [4] but also in fields such as economics, epidemiology, conservation, urban planning, and resource management [5–8].

The foundational work of renowned ecologists and biologists such as Simon A. Levin, David Tilman, and Mark E. J. Newman highlights the critical importance of the logistic growth model in population dynamics. Levin's research in ecological and evolutionary dynamics emphasized that social-ecological systems are complex adaptive systems in which processes at various scales interact and influence each other [9–14]. Tilman's studies on plant competition and biodiversity employed logistic growth models to reveal the underlying mechanisms that drive species coexistence and ecosystem stability [15]. Newman's contributions to statistical physics and complex systems have extended the applicability of logistic growth models to broader contexts, demonstrating their utility in understanding the growth and spread of populations in interconnected networks [16]. The enduring relevance of these pioneering works lies in their ability to provide a robust mathematical framework for exploring how populations grow, compete, and stabilize over time. For biologists and other researchers, the insights derived from these studies offer valuable tools for predicting and managing population dynamics across diverse ecological and evolutionary scenarios. Moving forward, leveraging the logistic growth model with contemporary data and advanced computational techniques would enable researchers to address pressing biological and ecological challenges with greater precision and predictive capacity.

One crucial component of the logistic growth model is the relationship between reproductive fitness and population density. In this model, the intrinsic growth rate captures the relationship between individuals' reproductive capacity and the population's growth rate. In classical logistic models, this biological metric is assumed to be constant, implying that individuals maintain an invariant reproductive capacity, unaffected by environmental conditions in which they reside. Various mathematical generalizations of the classical model aim to provide more flexibility through different mathematical formulations [17,18]. These formulations usually rely on phenomenological approaches to enhance the model's versatility, often incorporating parameters lacking biological meaning. Another model incorporating the link between population density and reproductive fitness is the well-known Allee effect [19,20], which assumes an inverse relationship between density and population growth once the population crosses a minimum size threshold.

The previously mentioned modeling approaches address density-dependent growth by incorporating attributes observed at the population scale. However, they often overlook individual-level reproductive and behavioral trade-offs, collectively influencing population dynamics. As a result, these models fail to provide insights into how information is transferred across scales [9,11,12,14]. Feedback mechanisms, such as how individual reproductive success can affect population density or how changes in population size can influence individual behavior, are crucial for understanding population dynamics.

Previous studies have extended population growth models by incorporating different approaches, including intraspecific interactions, spatial structure, environmental heterogeneity, demographic stochasticity, and resource availability [21–25]. Although these approaches produce variations in population growth rates, they do not explicitly model the incentives driving individual-level behavioral adaptation and their impact on population-level dynamics.

In reality, individuals' survival and reproductive abilities are expected to evolve dynamically in response to population size, which is influenced by self-decisions. Variation in individual traits, such as energy consumption and physical and health indicators, affects these decisions, directly impacting population growth. This has been observed in theoretical [26,27] and empirical population studies [28,29]. In addition to trait variation, individuals' decisions can be affected by spatiotemporal variation in the availability and quality of the resources individuals need to survive [30]. This variability in resources when foraging has been modeled in a consumer-resource context with different Holling functional responses to consumption [31,32], or through more complex models displaying different mechanisms of foraging such as body-size dependent differences in consumption rates [33] or resource vulnerability to consumption [34].

The connection between individual decision-making and ecological dynamics lies at the core of behavioral ecology [35–39]. Various modeling frameworks have been used to address the central challenge of understanding how collective dynamics emerge from individual decisions. Life history theories [40–43], theory of reproductive strategies [44,45], social foraging theory [46–50], and Optimal Foraging Theory (OFT) [51–53] constitute a robust theoretical foundation for studying individual foraging decisions. However, these frameworks are often not well suited to describing fine-scale, short-term behavioral traits that depend on environmental conditions, such as predation and starvation risks. In their pioneering work, Clark and Mangel introduced a modeling framework that approaches behavior from an evolutionary perspective within the context of an individual's life history, using a dynamic state-variable approach. This framework has been successfully applied to a wide range of ecological questions, including group size in social carnivores, host selection, clutch size in parasitic insects, seasonal survival strategies, and avian migration, among other paradigms [53–55]. In this study, we build on this foundational framework and couple it with the logistic model to study the trade-off between individual-level reproductive and behavioral traits and their influence on population growth.

Reconciling individual-level incentives with population-level dynamics requires a framework that explicitly incorporates the intertwined dynamics between population growth and individual decision-making processes that modulate reproductive fitness. Thus, understanding population-level features such as robustness and resilience requires characterizing the incentives and feedback mechanisms associated with individual behaviors.

This study considers population dynamics as a complex adaptive system in which individual-level decisions and population-level characteristics are interconnected. We model individuals' daily behavioral choices based on the overall system status, and as a result, the system continuously evolves in response to the updated individual-level behavioral decisions. We present an adaptive logistic growth model mainly inspired by the pioneering work of Clark and Mangel [53–55]. In our formulation, individuals make daily behavioral decisions about how much effort to invest in finding food for survival, considering their life history, current environmental conditions, and population characteristics. The decision-making process is mechanistically modeled in response to changes in the population characteristics, and mathematically formalized as a Markov Decision Process, focusing on two factors: (*i*) the energy gained if food is found, and (*ii*) the probability of finding food. The daily decision-making process poses a dynamic optimization problem centered on determining the optimal allocation of time and energy for searching for resources necessary for survival, henceforth referred to as "foraging time". Both factors shaping the continuous decision-making process dynamically influence the individuals' reproductive fitness and ultimately modulate the population's intrinsic growth rate. For instance, we might expect that the probability of finding

food per unit of foraging time decreases with larger population sizes, as resource acquisition becomes more competitive due to intraspecific competition.

Consequently, the population growth rate emerges from the aggregation of individual behavioral processes, which balance the cost-benefit trade-offs associated with the time spent foraging to secure resources for survival and reproduction. This methodology enables us to explicitly incorporate individual-level characteristics, such as varying levels of risk sensitivity, which affect foraging time choices. It also includes how differences in environmental quality impact foraging efforts required to attain similar population sizes. In this approach, population dynamics emerge from individuals' smaller-scale foraging processes. In contrast to the classic formulation, the intrinsic growth of the population is continuously updated depending on population size. However, this update is non-centralized and arises from purely individualized dynamic optimization processes.

In the Methods section, we present the mathematical formulation of our approach. The Results section explores key insights obtained with this approach and contrasts them with those from the classical approach. Our results suggest that populations with different individual-level characteristics might exhibit different population-growth trajectories, which are linked to varying effort levels made by the constituent individuals to support the exhibited population growth rate. Finally, in the Discussion section, we examine broader implications of our approach, emphasizing the importance of incorporating individual behavioral trade-off processes that drive population-scale dynamics.

## Materials and methods

In our adaptive behavior approach, we explicitly model the decision-making processes of individuals and their impact on population dynamics. The population is assumed to be a complex adaptive system in which each individual's behavior–specifically, their daily foraging time–affects reproductive fitness and, consequently, population growth. The intrinsic reproduction rate is not fixed; instead, it is dynamically modulated by the individuals' daily energy levels, which results from their behavioral choices. To formalize this, we define the daily intrinsic population growth rate at time $t$ as $r_t = r_c \cdot e_t$, where $r_c$ is a constant linking individual energy to reproductive contribution, and $e_t$ denotes the daily energy level of a representative individual. This coupling bridges micro-level decision-making with macro-level population growth. At each time step, the representative individual selects the total daily foraging time spent searching for food. The chosen behavioral decision balances potential immediate foraging cost and energy gains, as well as discounted expected cost-benefit tradeoffs over the planning horizon, the period during which a representative individual evaluates the cost-benefit trade-offs that guide its behavioral choices. Therefore, in this framework, the logistic growth equation is iteratively evaluated with an adjusted intrinsic growth $r_t$, which is associated with the individuals' daily energy $e_t$. The solution incorporates the time-varying population size ($P(t)$), its associated population growth rate ($r_t$), and the corresponding individuals' expected energy state ($e_t$).

We use a mean-field formulation to model population growth and a Markov decision framework to model adaptive behavior as used in [56–63]. Behavioral changes are represented as adjustments in individuals' foraging time within an acceptable range $f \in \mathcal{F} = [f_{\min}, f_{\max}]$. These adjustments aim to maximize net energy gains while accounting for energy consumption. The decision-making process is forward-looking, considering the current and future potential states of finding and not finding food, denoted by $\mathcal{S} = \{FF, NFF\}$. We assume that foraging individuals obtain energy based on the functional form $u(f) = \alpha^h (b^h f - f^2)^\nu$ where $f$ denotes the foraging time at day $t$. Notice that $f_{\text{opt}} = b/2$ is the foraging time selection that

yields the maximum energy gain, assuming that food is found at this foraging time selection. We incorporate the normalization parameter $\alpha^h$ to maintain the utility values within a given range.

The utility function yields the net energy gain for each unit of foraging time. Its concave shape indicates that increasing the time spent foraging initially leads to higher energy gains. However, beyond a certain point, the additional effort required to spend more time searching for food increases energy expenditure, reducing the net gain. The parameter $\nu$ controls the slope of the utility function and we use it to model the population's sensitivity to environmental conditions. A higher value of $\nu$ results in steeper utility curves, which could be interpreted as reflecting individuals with a lower sensitivity or a reduced propensity to adjust their optimal foraging time, since doing so may lead to greater energy losses (Fig 1). We assume that each foraging time choice $f$ is associated with a probability of finding food denoted by $P^{FF}(f, P(t))$, which also depends on the population size at time $t$, $P(t)$. Intuitively, this probability should increase with $f$, as spending more time foraging is expected to raise the likelihood of finding food. However, it should also decrease when the population increases, since greater competition may reduce the availability of resources. Notice that there is not a single way to model these dynamics mathematically. In the SI appendix, we present an example of a possible formulation for the $P^{FF}(f, P(t))$ function and explore the significant effect that its specific shape can have on the overall model outcomes.

The Markov decision process incorporates expected utilities, which inform the individuals of the net utility to gain over a planning horizon ($\tau$). For each possibility, we compute the corresponding value functions—or expected utilities—$V_t^{FF}$ and $V_t^{NFF}$ for each time step during the planning horizon, $[t, t + \tau]$. The expected utility of finding food is obtained using the following Bellman equation:

$$V_t^{FF} = \max_{f \in \mathcal{F}} \left\{ u^{FF}(f) + \delta \left[ P^{FF}(f) V_{t+1}^{FF} + (1 - P^{FF}(f)) V_{t+1}^{NFF} \right] \right\}$$

(1)

$$\text{for} \quad t = 1, 2, \cdots, \tau - 1.$$

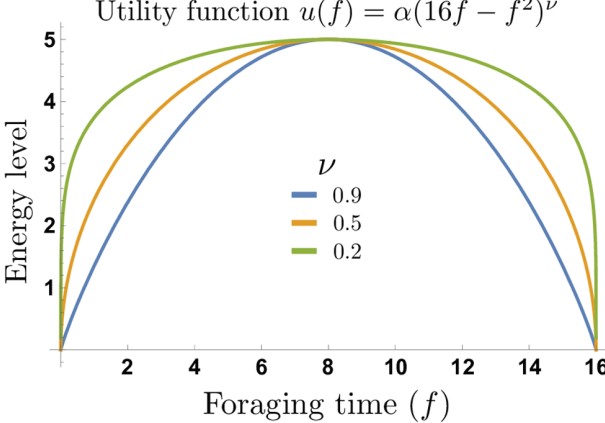

**Fig 1. Energy gain as a function of foraging time.** We assume that the agent has an optimal foraging time $f_{\text{opt}} = 8$, for scenarios of sensitivity $\nu = \{0.9, 0.5, 0.2\}$. The parameter $\alpha$ normalizes the maximum energy value across sensitivity scenarios $u(f_{\text{opt}}) = e_{max}$.

Eq (1) formalizes the search for the optimal foraging selection by considering the immediate utility $u^{FF}(f)$ and discounting the future expected utility by a discount factor $\delta$. The expected utility of not finding food at time $t$, denoted by $V_t^{NFF}$, is similarly computed using the Bellman equation

$$V_t^{NFF} = \max_{f \in \mathcal{F}} \left\{ u^{NFF}(f) + \delta \left[ P^{FF}(f) V_{t+1}^{FF} + (1 - P^{FF}(f)) V_{t+1}^{NFF} \right] \right\} \tag{2}$$

$$\text{for} \quad t = 1, 2, \cdots, \tau - 1. \tag{3}$$

In this framework, the decision-making process determines the optimal foraging choice $f_{\text{opt}}$ at each time-step of the planning horizon $[t, t + \tau]$. The daily optimal selection is obtained by evaluating the expected utilities associated with each possible foraging time $f$, taking into account both the immediate utility $u^{FF}(f)$ and the discounted future utilities of either finding food $V_{t+1}^{FF}$ or not finding food $V_{t+1}^{NFF}$. This optimization balances short-term gains with long-term outcomes, leading the agent toward foraging decisions that maximize overall net utility across the planning horizon.

Equations (1) and (2) are solved using a backward induction approach. The procedure assumes that at the end of the planning horizon $V_\tau^{FF} = u(f_{\text{opt}})$, representing the utility at the immediate optimal foraging time, and $V_\tau^{NFF} = 0$. The optimal foraging time selection is then computed as the result of the final iteration, and the corresponding daily energy gain is given by the value of the immediate utility at the selected optimal foraging time: $e = u^{FF}(f_{\text{opt}})$.

## Results

We first analyze how adaptive foraging behavior influences population growth dynamics, comparing our adaptive logistic model results to those from the classical logistic model (Figs 2 and 3). Adaptive behavior prompts individuals to modify their daily foraging time based on population density. As the population size increases, individuals' optimal foraging time increases due to intensified resource competition (Fig 2B). Eventually, the optimal decision stabilizes above the initially optimal foraging time ($f_{\text{opt}} = 14$), reflecting greater foraging effort required to sustain adequate energy levels for survival and reproduction (Fig 2C).

When individuals display lower sensitivity to population changes (higher $\nu$), the deviations from the optimal foraging time are significantly reduced, enabling populations to sustain similar sizes with comparatively less foraging effort (Fig 3) This indicates that populations with lower behavioral sensitivity (i.e., less reactive to environmental or social cues) are less affected by environmental changes because their life history traits reduce their dependency on immediate conditions. As a result, they are more adaptable to density-induced resource limitations.

Additionally, we characterize the trade-off between foraging effort and energy gain as populations reach their carrying capacity. Populations with higher sensitivity (lower $\nu$ values) ultimately settle into higher foraging times and lower energy levels, reflecting greater energetic costs due to competition (Fig 4) Conversely, populations with lower sensitivity maintain higher energy levels and lower foraging effort.

Comparing the adaptive and classical logistic models, we observe attenuated growth trajectories under adaptive foraging. To compare population growth dynamics between the adaptive and classic modeling frameworks, we use the maximum population difference, defined as the absolute value of the maximum difference between the population trajectories of the adaptive and the classic models: $\max(|P(t) - P_0(t)|)$, where $P(t)$ is the population size of the adaptive model, and $P_0(t)$ is the population size of the classic model, computed using $r_t = r_c * u(f_{\text{opt}})$. We find that the most significant deviations occur at intermediate levels of

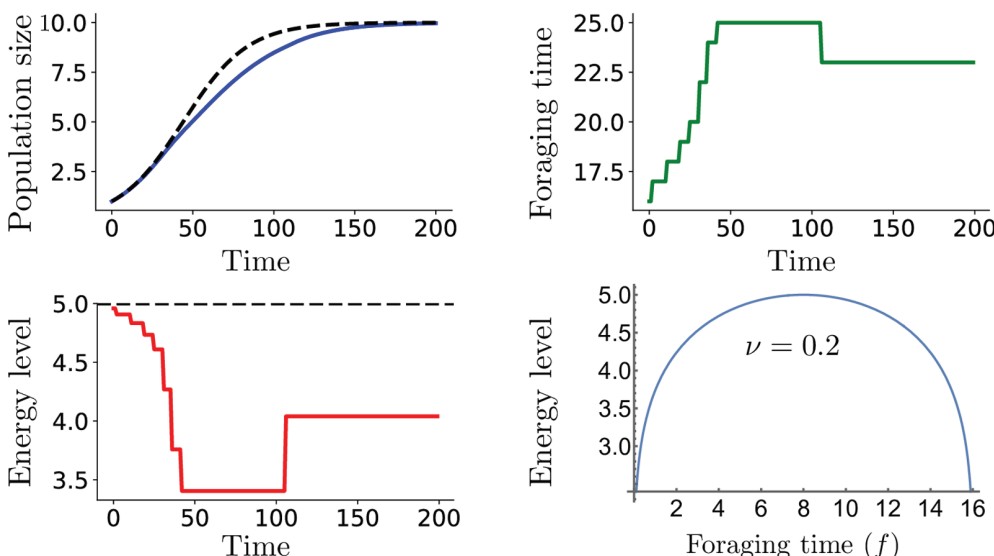

**Fig 2. Highly sensitive populations require greater foraging effort.** Population growth, optimal daily foraging time, and net daily energy gain are shown for an adaptive logistic growth model. Higher sensitivity increases resource competition, requiring longer foraging times to maintain sufficient energy levels. Parameters $\nu = 0.2, \tau = 7, \delta = 0.98, k = 10$, and $r_c = 0.01$.

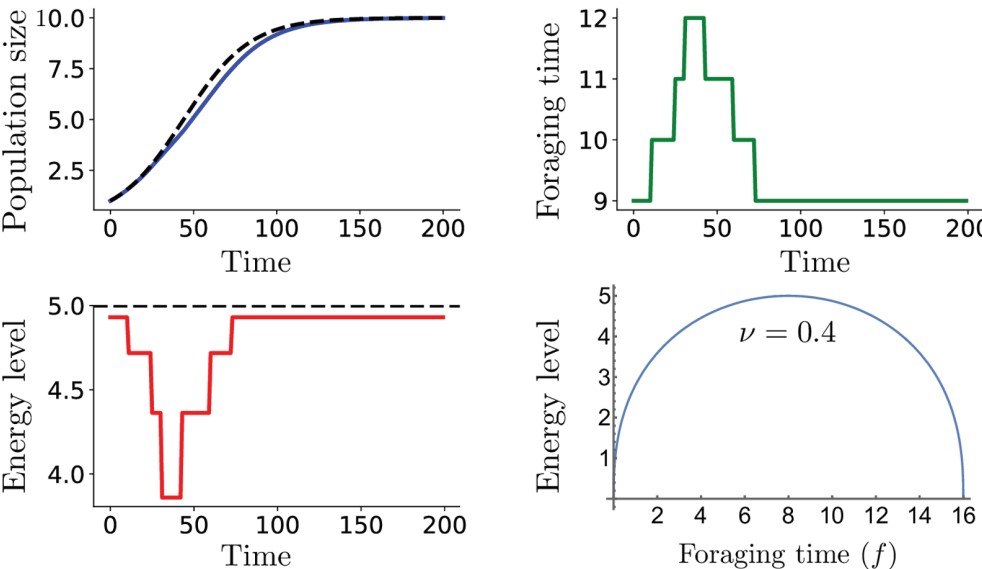

**Fig 3. Highly sensitive populations require greater foraging effort.** Population growth, optimal daily foraging time, and net daily energy gain are shown for an adaptive logistic growth model. Reduced sensitivity leads to lower resource competition, allowing populations to sustain adequate energy levels with less foraging effort. Parameters $\nu = 0.4, \tau = 7, \delta = 0.998, k = 10$ and $r_c = 0.01$.

the population limiting size (Fig 5) Highly sensitive populations quickly adapt their foraging time, experiencing noticeable energy losses. In contrast, populations with very low sensitivity show minimal deviation from classical growth, as their behavior remains unchanged.

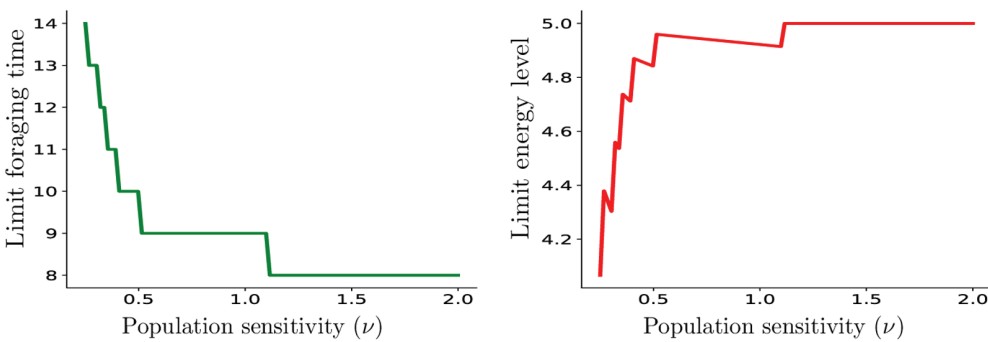

**Fig 4. Limit Foraging Times and Energies depending on the sensitivity parameter $\nu$.** Populations with higher sensitivity exhibit longer foraging times and lower steady-state energy levels, reflecting greater energetic costs from competition. In contrast, lower-sensitivity populations maintain higher energy and reduced foraging effort. Parameters $\tau = 7$, $\delta = 0.98$, $b = 16$, $k = 10$ and $r_c = 0.01$, and a utility function that has a maximum value of $u = 5$ in all cases, attained at $f_{opt} = 8$.

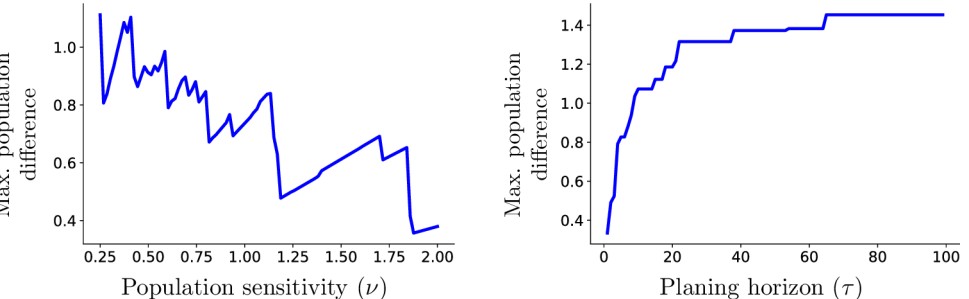

**Fig 5. Maximum population difference between classical logistic constant and the adaptive logistic formulation.** (Left panel) The difference decreases as the sensitivity parameter $\nu$ increases; (Right panel) The difference increases as the planning horizon $\tau$ extends. Parameters $\delta = 0.98$, $b = 16$, $k = 10$, $r_c = 0.01$, a value of $\tau = 14$ for the left panel and a value of $\nu = 0.5$ for the right.

These results highlight how individual-level behavioral trade-offs dynamically influence population-level outcomes, emphasizing the importance of integrating adaptive behaviors into ecological modeling frameworks.

In this framework, foraging time selections and energy levels vary throughout population growth due to these differences in sensitivity. The first, most sensitive population is forced to increase its foraging effort due to its heightened responsiveness to changes in conditions, which reduces its energy levels. In contrast, the second, more elastic population can return to the optimal foraging decision and maintain its energy levels. Although both populations reach the same total capacity within similar time frames, the first does so with significantly greater foraging effort than the second.

Fig 4 further illustrates the trade-off between the agents' foraging effort and energy gain as the population approaches its carrying capacity–that is, the limiting values of foraging time and energy level.

In addition to providing a more detailed characterization of individuals' energy state and behavioral choices, the adaptive framework yields attenuated population growth curves compared to the classic model. The reasoning is that as individuals adjust their foraging time selections in response to environmental pressures, they deviate from their a priori

optimal energy gain. Consequently, they experience energy losses driven by the increasing competition for food, which constrains their decision-making rationale and reduces overall growth.

Finally, for highly sensitive agents (*i.e.* low values of $\nu$), the population growth trajectory for the adaptive framework does not differ substantially from that of the classical approach. In these cases, the utility function is almost flat, resulting in low marginal benefits from increasing foraging time. It follows that there is no significant energy gain across foraging time options. Conversely, the difference between the classic and adaptive logistic models becomes negligible for low-sensitive populations (*i.e.* high values of $\nu$). In these scenarios, individuals do not perceive enough incentives to adapt their behavior, see Fig 5. The adaptive framework captures individual-level behavioral trade-offs, illustrating how aggregated decision-making can influence population growth dynamics.

## Discussion and conclusion

Our modeling framework builds upon the classic logistic model to highlight the interplay between individual-level incentives and population-scale dynamics. By integrating optimal foraging into the logistic model, we capture the dynamic interplay between individual decision-making and population-scale processes in greater detail. Our findings highlight the importance of incorporating individual adaptive behaviors into ecological models. Traditional logistic growth models, which assume a constant intrinsic growth rate, overlook the dynamic nature of individual behavior as population size increases. As competition intensifies, individuals must invest more effort in foraging to obtain the same resources and maintain their energy levels. By incorporating individuals' behavioral adjustments, our adaptive framework offers a more realistic depiction of population dynamics. Our modeling approach captures the influence of resource availability, competition, and individual health on reproductive fitness and population growth.

These results are consistent with empirical observations of migratory species that rely on resources that provide low energy per individual unit. In such cases, like that of leatherback turtles, the additional energy expended migrating to areas of high prey density is offset by the benefits, even if it results in reduced reproductive growth [64]. This trade-off does not appear in species with sufficiently high fat reserves to sustain their reproductive efforts [65]. Other examples of adaptive behaviors that involve balancing individual traits include: the activity patterns of zebrafish (Danio rerio) which exhibit lower movement rates in dark environments and higher rates in illuminated ones [66]; insects decisions when selecting feeding or oviposition sites [55]; and the group hunting strategies and habitat use observed in African lions (Panther leo) [67,68].

The adaptive framework shows that as population size increases, individuals must exert more effort in finding food. This results in higher energy expenditure and reduced net energy gain, ultimately affecting population growth. Our findings suggest that the best foraging time increases as the population grows, reflecting the intensified competition for limited resources. Furthermore, as the population grows, individuals may be displaced to less favorable foraging areas, where increased competition with other populations or worse environmental conditions may lead to even higher energy expenditures during foraging [69]. This finding supports the notion that higher population densities reduce per capita resource availability, requiring behavioral adjustments for survival. The framework enables the modeling of different fitness characterizations through the shape of the utility function, which captures the inherent trade-off between foraging time effort and energy gain, influencing the individuals' adaptation to a niche. This flexibility allows the model to capture a broader range of ecological scenarios and

population attributes; however, it also increases the complexity and cost of accurate modeling. Our numerical experiments in the SI Appendix highlight this feature, particularly regarding the choice of the probability of finding food and the utility function. The scenarios presented in the results section show that distinct characterizations of the strain or burden placed on the population to survive become evident in this modeling framework. This aspect remains hidden when considering only the evolution of population growth.

Additionally, our study offers insights into individual energy dynamics in relation to their sensitivity to environmental changes. Populations with higher sensitivity (i.e., lower $\nu$ values) exhibited increased foraging times and energy expenditures. In contrast, populations with lower sensitivity (higher $\nu$ values) could maintain optimal energy levels with minimal deviation from their optimal foraging times. This highlights the adaptive framework's ability to capture individuals' nuanced trade-offs to balance energy intake and expenditure in response to population density and resource competition.

The adaptive framework quantitatively measures the strain or burden placed on individuals within a population. Examining the foraging time choices and energy levels can help assess the effort required for survival and reproduction under different population densities. Hence, this approach enables the identification of incentive regimes that drive behavioral responses at the individual scale, whose aggregation significantly influences population-scale dynamics [14]. This level of modeling granularity provides a deeper understanding of how individual contributions shape population growth and persistence by providing insights into the nature of life history tradeoffs.

Integrating an individual-level behavioral model with population dynamics provides a more comprehensive understanding of individuals' behavioral choices and their implications across scales. Our findings suggest that the resilience of a population is not an inherent characteristic of the population as a whole, but rather a result of individual-level decisions shaped by their life history tradeoffs [70]. As individuals modify their foraging behavior in response to competition and population density, the population's capacity to sustain itself and grow depends mainly on these individual decisions. Consequently, population resilience emerges from the adaptive strategies individuals adopt in response to a dynamic and competitive environment. Viewing resilience as an emergent property of individual-level adaptations offers valuable insights for managing and conserving populations across diverse ecosystems. This modeling perspective highlights the need for integrated and robust conservation policies that account for environmental factors and species' dynamic behavioral responses to changing conditions.

The adaptability of the framework used in this method is a significant result of our study. Our approach using dynamic programming for individual optimal decision-making has proven valuable in incorporating individual-level decision processes while integrating them with global population dynamics. This framework has previously been applied to study the impact of individual-level incentives driving adaptive human behavior on the dynamics of infectious diseases [56,57,59–63]. In this article, we expand its application to enhance logistic growth processes. However, the fundamental concept can be applied to other methods where individual-level decisions influence overall system dynamics. Integrating individual-level behavioral models with population-scale dynamics offers a more comprehensive and realistic understanding of population growth. By accounting for individual decision-making processes and their influence on reproductive fitness, our adaptive logistic growth model provides valuable insights into the complex interplay between individual actions and population dynamics. Finally, this approach enhances our understanding of population dynamics and has practical applications in ecology, resource management, and conservation research. It offers a robust

framework for predicting and managing population dynamics in the face of environmental challenges.

## Supporting Information

**S1 File. Supporting information file.** Sensitivity of the probability of finding food and the utility function.
(PDF)

## Author contributions

**Conceptualization:** Jimmy Calvo-Monge, Baltazar Espinoza, Fabio Sanchez.

**Formal analysis:** Jimmy Calvo-Monge.

**Investigation:** Jimmy Calvo-Monge, Baltazar Espinoza, Fabio Sanchez, Jorge Arroyo-Esquivel.

**Methodology:** Jimmy Calvo-Monge, Baltazar Espinoza, Fabio Sanchez.

**Supervision:** Baltazar Espinoza, Fabio Sanchez.

**Validation:** Jimmy Calvo-Monge, Baltazar Espinoza, Fabio Sanchez, Jorge Arroyo-Esquivel.

**Writing – original draft:** Jimmy Calvo-Monge, Baltazar Espinoza, Fabio Sanchez.

**Writing – review & editing:** Jimmy Calvo-Monge, Baltazar Espinoza, Fabio Sanchez, Jorge Arroyo-Esquivel.

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
