## [Decision Letter · Decision Letter 0]

21 Mar 2025

PONE-D-25-06482The interplay between foraging choices and population growth dynamicsPLOS ONE

Dear Dr. Sanchez,

Thank you for submitting your manuscript to PLOS ONE. After careful consideration, we feel that it has merit but does not fully meet PLOS ONE’s publication criteria as it currently stands. Therefore, we invite you to submit a revised version of the manuscript that addresses the points raised during the review process.

**This is a timely work with a new model for combining foraging and theoretical logistic growth models to elucidate the dynamics of populations, particularly individual foraging decisions and the impacts on growth. This would be of broad encompassing utility across vast biological systems.However, what clearly stands out is the lack of in-depth ecological analyses demonstrating applicable or transferable knowledge across some key populations existing within families of the taxonomic kingdoms. In alignment with the reviewers, more ecological research on foraging behaviour should rightly be featured in this study for the wider benefit of ecologists. Also, I strongly recommend the manuscript is edited for English quality, beginning from the abstract.** **Please submit your revised manuscript by May 05 2025 11:59PM. If you will need more time than this to complete your revisions, please reply to this message or contact the journal office at plosone@plos.org. **

**Please include the following items when submitting your revised manuscript:**

**A rebuttal letter that responds to each point raised by the academic editor and reviewer(s). You should upload this letter as a separate file labeled 'Response to Reviewers'.****A marked-up copy of your manuscript that highlights changes made to the original version. You should upload this as a separate file labeled 'Revised Manuscript with Track Changes'.****An unmarked version of your revised paper without tracked changes. You should upload this as a separate file labeled 'Manuscript'.**

****

We look forward to receiving your revised manuscript.

**Kind regards,**

**Charlotte Chibuzor Ndiribe, Ph.D.,**

**Academic Editor**

**PLOS ONE**

 [National Science Foundation (NSF) through Expeditions in Computing Grant CCF-1918656

NSF through Incorporating Human Behavior in Epidemiological Models (IHBEM) grant DMS-2327710]. 

4. We note that your Data Availability Statement is currently as follows: [All relevant data are within the manuscript and its Supporting Information files.

**Additional Editor Comments >**

**Abstract:**

**“... the logistic growth model to account for dynamic incentives depending on [delete the] population size. Our results yield insights into the multiscale processes, such as __ and ___,that impact population robustness beyond mere size and aggregated reproductive traits. **

**This variation may be induced by environments imposing distinct behavioral cost-benefit trade-offs that require individuals to exert distinct levels of [efforts on what?]**

**Introduction:**

**Line 17: Remove “has” emphasised.**

**Line 19: Remove “have” utilised.**

**Line 21: Similar tense issue.**

**Line 24: “lie in”.**

**Line 35: With a bracket explain agent, few words.**

**Line 49: “insights into” or “on information transfer across scales”.**

**Others: Line 52, 60, 68 with “envision”, 78 “time effort”?, 79 “individual-level processes”, 82 delete “the”, many others with “the” issues. Line 97, this is not a proposal, Line 102-110 rewrite to meet contemporary standards.**

**Methods:**

**Write early sentences and first paragraph well. Clearly distinguish Sampling strategy from the statistical development.**

**Line 136: “We may use” Rephrase, please.**

**Results:**

**The first sentence begs for clarity, what are you referring to? Remove unnecessary words/ statements here and throughout this work. **

**Rewrite results section with proper captions, other than figures. Figures only depict a set of tests, those are your captions. Again, it is important to adhere to modern styles of writing science. Rightly, tables and figures are bracketed.**

**Discussions:**

**You need test organisms or about 2-3 study systems as references to support and validate the utility of your model as claimed. Again, check and correct all grammatical lapses here.**

**Conclusions:**

**This should not start again with a repeat “In conclusion”.**

****

**Reviewers' comments:**

**Reviewer's Responses to Questions**

**Comments to the Author**

**1. Is the manuscript technically sound, and do the data support the conclusions?**

**The manuscript must describe a technically sound piece of scientific research with data that supports the conclusions. Experiments must have been conducted rigorously, with appropriate controls, replication, and sample sizes. The conclusions must be drawn appropriately based on the data presented. **

**Reviewer #1: Yes**

**Reviewer #2: Yes**

**2. Has the statistical analysis been performed appropriately and rigorously? **

**Reviewer #1: I Don't Know**

**Reviewer #2: Yes**

**3. Have the authors made all data underlying the findings in their manuscript fully available?**

**The PLOS Data policy requires authors to make all data underlying the findings described in their manuscript fully available without restriction, with rare exception (please refer to the Data Availability Statement in the manuscript PDF file). The data should be provided as part of the manuscript or its supporting information, or deposited to a public repository. For example, in addition to summary statistics, the data points behind means, medians and variance measures should be available. If there are restrictions on publicly sharing data—e.g. participant privacy or use of data from a third party—those must be specified.**

**Reviewer #1: Yes**

**Reviewer #2: Yes**

**4. Is the manuscript presented in an intelligible fashion and written in standard English?**

**PLOS ONE does not copyedit accepted manuscripts, so the language in submitted articles must be clear, correct, and unambiguous. Any typographical or grammatical errors should be corrected at revision, so please note any specific errors here.**

**Reviewer #1: Yes**

**Reviewer #2: Yes**

**5. Review Comments to the Author**

**Please use the space provided to explain your answers to the questions above. You may also include additional comments for the author, including concerns about dual publication, research ethics, or publication ethics. (Please upload your review as an attachment if it exceeds 20,000 characters)**

**Reviewer #1: Fabio Sanchez and his collaborators explore theoretically the interplay between individual-level behavior and population growth dynamics. Although the text and analyses look solid and convincing, I realized that my (limited) background in theoretical ecology does not allow me to thoroughly evaluate the paper and, in particular, the models presented.**

**However, one problem I noticed is that the authors do not discuss much how their modeling framework could be used in real biological systems. I would appreciate if some examples were given (I think this modeling framework could be used for many different systems, including plant or insect populations). It could also be emphasized how the models should be parameterized for different systems.**

**Reviewer #2: Overview**

**Sanchez et al. have developed a model that integrates a foraging model into the logistic growth model. This is an interesting model that provides some novel insights into the impact of the ‘average’ individual-level response on population growth. I believe the model developed is most certainly worthy of publication, but I feel the authors have overlooked a wealth of ecological research on foraging behaviour that would greatly benefit, provide insights to the practitioner of how this model could be used, and the ecological literature could perhaps inform the model parameterization.**

General Comments

The model develop is very interesting and I think adds to the general ecological modelling literature. However, as noted above, there is a wealth of literature related to foraging behaviour that is not discussed in any level of detail here and I think would benefit the paper. Putting this model in the context of what has been already done will enhance the relevance and impact of this paper on the ecological literature. For example, the shape of the foraging function and the probability of finding food which underlie the model could be based on different species life-histories, such as ambush predators, generalist predators, grazing predators, scavengers, etc. By doing this, the impact of this method on the population growth rates of different ‘types’ of life-histories could be captured. In classical ecological literature, the idea of a functional response is well established and should be discussed whenever talking about modelling foraging behaviour (e.g. Holing, https://esajournals.onlinelibrary.wiley.com/doi/pdf/10.1890/0012-9623-95.3.200).

Also, in the marine realm ideas around foraging arena theory likely has relevance to this work. (see a number of Carl Walters papers or https://pressbooks.bccampus.ca/ewemodel/chapter/foraging-arena-theory/).

Specific Comments

Line 168: It is not clear to me what the ‘planning horizon’ represents (also give it the symbol tau). As a result, on

line 234, what planning effort means, in ecological terms, is unclear.

Lines 184-192: I think it would be useful to show the utility function underlying the models for Figure 2 and 3 in the main text. I would suggest showing these two utility functions in Figure 1.

Line 186: replace “stale’ with “stable”.

Lines 200-218 are more of a discussion that results.

Line 200-201: Simplify this to “Figs 2 and 3 show how this framework visualizes processes….”

Line 205: “Resilience” in ecological terms generally is related to the ability of a population to recover from low abundance, or to sustain itself in the face of some detrimental change in the environment, so I would avoid that word here. It is more that the impact of this ‘life history’ on the population growth is less that the impact of the life history in Figure 2.

Line 213: Rather than ‘energy drain’, I would suggest ‘foraging effort’.

**For the supplemental section, it would be nice in the methods to link to the relevant sections of the methods.**

**6. PLOS authors have the option to publish the peer review history of their article (what does this mean?). If published, this will include your full peer review and any attached files.**

**Reviewer #1: No**

**Reviewer #2: No**

****

**While revising your submission, please upload your figure files to the Preflight Analysis and Conversion Engine (PACE) digital diagnostic tool, https://pacev2.apexcovantage.com/. PACE helps ensure that figures meet PLOS requirements. To use PACE, you must first register as a user. Registration is free. Then, login and navigate to the UPLOAD tab, where you will find detailed instructions on how to use the tool. If you encounter any issues or have any questions when using PACE, please email PLOS at figures@plos.org. Please note that Supporting Information files do not need this step.**

---

## [Author Response · Author response to Decision Letter 1]

13 May 2025

We have attached a response letter addressing all the reviewer's and editor's comments.

---

## [Editor Report · Decision Letter 1]

21 May 2025

The interplay between foraging choices and population growth dynamics

PONE-D-25-06482R1

Dear Dr. Espinoza,

We’re pleased to inform you that your manuscript has been judged scientifically suitable for publication and will be formally accepted for publication once it meets all outstanding technical requirements.

Kind regards,

Charlotte Ndiribe, Ph.D.

Academic Editor

PLOS ONE

Additional Editor Comments (optional):

Dear Dr. Espinoza et al.,

Thank you for your insightful contribution to the Plos ONE Journal.
---

## [Editor Report · Acceptance letter]

PONE-D-25-06482R1

PLOS ONE

Dear Dr. Espinoza,

I'm pleased to inform you that your manuscript has been deemed suitable for publication in PLOS ONE. Congratulations! Your manuscript is now being handed over to our production team.

Kind regards,

on behalf of

Dr. Charlotte Ndiribe

Academic Editor

PLOS ONE